# Robotic Platform for Horticulture: Assessment Methodology and Increasing the Level of Autonomy

**DOI:** 10.3390/s22228901

**Published:** 2022-11-17

**Authors:** Alexey Kutyrev, Nikolay Kiktev, Marcin Jewiarz, Dmitriy Khort, Igor Smirnov, Valeria Zubina, Taras Hutsol, Marcin Tomasik, Mykola Biliuk

**Affiliations:** 1Federal State Budgetary Scientific Institution “Federal Scientific Agroengineering Center VIM”, International Center of Informatics and Computer Science (ICICS), Hong Kong 109428, China; 2Department of Intelligent Technologies, Taras Shevchenko National University of Kyiv, 01601 Kyiv, Ukraine; 3Department of Automation and Robotic Systems, National University of Life and Environmental Sciences of Ukraine, 03041 Kyiv, Ukraine; 4Faculty of Production and Power Engineering, University of Agriculture in Krakow, 30-149 Krakow, Poland; 5Department of Mechanics and Agroecosystems Engineering, Polissia National University, 10008 Zhytomyr, Ukraine; 6Innovative Program of Strategic Development of the University, European Social Fund, University of Agriculture in Krakow, 30-149 Krakow, Poland

**Keywords:** robotization of agriculture, evaluation factors, experts, robot autonomy, control system architecture, trajectory of movement, software module for route construction

## Abstract

The relevance of the study is confirmed by the rapid development of automation in agriculture, in particular, horticulture; the lack of methodological developments to assess the effectiveness of the introduction of robotic technologies; and the need to expand the functionality of mobile robots. The purpose of the study was to increase the level of autonomy of a robotic platform for picking apple fruits based on a new method, develop a system of factors to determine the effectiveness of the introduction of robots in horticulture, and develop a control system using integrated processing of onboard data. The article discussed the efficiency factors for the introduction of robotic systems and technologies in agricultural enterprises specializing in horticulture within the framework of projects with different budgets. The study sample consisted of 30 experts—enterprises that have implemented robotic platforms and scientists specializing in this field. Based on an expert survey of enterprise specialists, a ranked list of 18 efficiency factors was obtained. To select an evaluation factor that determines the effectiveness of robotization and the developed control system, a method for calculating the concordance coefficient (method of expert analysis) was applied as a measure of the consistency of a group of experts for each group of factors. An analysis of the results of the expert evaluation showed that three factors are the most significant: the degree of autonomy of work; positioning accuracy; and recognition accuracy. The generalized indicator of local autonomy of task performance was estimated based on the analysis of a set of single indicators. A system for controlling the movement of an autonomous robotic wheeled platform based on inertial and satellite navigation and calculation of the path to be overcome was developed. The developed software allows for the design of a route for the robotic platform in apple horticulture to automatically perform various technological operations, such as fertilization, growth and disease control, and fruit harvesting. With the help of the software module, the X, Y coordinates, speed and azimuth of movement were given, and the movement of the platform along the given typical turn trajectories in an intensive horticulture environment was visualized.

## 1. Introduction

The modern level of development of infocommunication and computer technologies, microprocessor technology and equipment, communication and positioning makes possible the development and practical application of automated and robotic technologies and technical means to improve the efficiency of agricultural production. Currently, intensive horticulture is becoming increasingly widespread due to rapid fruiting and high yield rates. At the same time, the process of harvesting apples in intensive horticulture is the most time-consuming, and harvesting is carried out mainly by a team of pickers. In the production process of cultivating fruit crops, this is an important final stage which requires the development of automated devices and robotic platforms with a control system capable of offline harvesting.

With the undoubted advantages of the known approaches to robotization of harvesting operations in gardens, the relationship between the indicator (degree) of autonomy of robots and the number of functions they implement, for example, performing various agricultural work on one robotic platform, has not been sufficiently studied. The creation of appropriate calculation methods will reveal the potential of expanding the functionality of mobile robots by increasing the degree of their autonomy. 

At the same time, the issues of increasing the technical efficiency of solutions aimed at achieving a high level of robot autonomy require development. This can be achieved by applying intelligent approaches to the complex processing of data coming from a complex of information devices.

The autonomy of a robotic platform is the ability to perform a technological operation in time, in space, in conditions of changing tasks, under changing environmental conditions without the need to interact with other subjects or subjects of the highest level of the hierarchy (Figure 1).

The expansion of the functionality of the robotic platform by increasing the level of its autonomy using integrated onboard data processing is an urgent scientific task, the solution of which will allow the technological operation of autonomous harvesting of apple fruit to be carried out qualitatively. 

The aim of this research was to increase the level of autonomy of the robotic platform for apple fruit harvesting based on a scientifically proven new method and to develop a control system using integrated onboard data processing.

## 2. Literature Review and Problem Statement

Mechanization of agriculture has significantly increased labor productivity. However, in many branches of agriculture, especially in horticulture, manual labor still accounts for up to 50% of costs [1,2,3]. In this regard, the development of robotic solutions for use in agriculture is actively developing. There are examples of commercial use of automated wheeled tractor equipment in the preparation and conduct of sowing operations, weed control and pest control, yield forecasting and harvesting of grain crops. The use of robots to automate gardening is still an actively developing area. Despite the fact that such robotic machines began to be created in the late 1960s, robots in gardening have not yet been brought to commercial use, although many prototypes have been developed [4,5]. In particular, there are prototypes of wheeled platforms for collecting fruits using manipulators. When moving in rows of horticulture plantings, the route is usually planned in advance by navigating through the aisles, and not by individual trees [6,7,8,9,10,11]. The platform presented by Australian researchers N. Shalal, T. Low, C. McCarthy and N. Hancock in [6], moved along a pre-designed map of the horticulture with correction based on laser scanning, and was able to move along the aisle and avoid obstacles. The platform presented by Northwest Nazarene University (USA) scientists A. Villemazet, A. Durand-Petiteville and V. Cadenat in article [7] implemented unmanned movement in horticulture using computer vision and ultrasonic sensors. Field tests in a peach horticulture with a length of 27 m and a width of 6.4 m showed that the RMS positioning error (RMSE) was 3.5 cm. A similar platform was presented by Chinese researchers Yan Song, Feiyang Xu, Qi Yao et al. in article [8]. Using 2D lidar processing using a particle filter (PF) and a Kalman filter (KF), field tests showed a positioning error of 5.5 cm for PF and 8.8 cm for KF. However, these platforms [7,8] cannot make turns between aisles in automatic unmanned mode. In article [9] by New Zealand researchers M.H. Jones, J. Bell, D. Dredge et al., a platform for moving containers through horticulture was presented. This GPS-based platform with four wheels and an independent control demonstrated 6.0 cm RMSE positioning during field tests. In the work of researchers Bayar G., Bergerman M., et al. [10], a robotic platform moved along the aisle using positioning based on laser scanning. As a result, the platform was able to move along the aisle, as well as from row to row. An interesting approach to planning the movement of a robotic platform in gardens based on the adaptation of the B-patterns approach, was presented by authors Bochtis D., Griepentrog H.W., Vougioukas S. et al. in [11]. In this case, the one with the greatest useful path was chosen as the optimal route.

The theory of automatic control of wheeled platforms began to appear from the moment of miniaturization of computing tools able to be installed on the platforms. One of the fundamental works in this field was the equations of motion proposed in 1981 in the work of the American researcher Mac-Adam, C.C. [12]. In the research of scientists Nenajdenko, A.S., Poddubnyj, V.I., Valekzhanin, A.I. a multifactorial model of differential equations of motion of wheeled vehicles along a complex curved trajectory was considered, which made its configuration and application for work in the horticulture quite time-consuming and requiring qualified specialists [13].

Researchers D. Khort, A. Kutyrev, N. Kiktev et al. [14,15,16,17,18], carried out the development and implementation of robotic platforms for agricultural production in the horticulture. In particular, the features of the developed robotic platform for harvesting strawberries [15], apples [16], processing plants with a solution in the form of hot mist [14] were described. The control system was based on an Arduino microcontroller and control software written in Python. The article [18] provided a theoretical calculation of the main design and technological parameters, describes electronic components and assemblies. Robotic platforms are versatile, simple in design, easily adapt to various working bodies and actuators, which is important for their use in various technological operations in the garden. Modeling of multi-agent robotic systems for horticulture robots based on pre-compiled scenarios was described in article [17]. The task of evaluating the robotic platform and increasing its level of autonomy remains unresolved.

In article [19], researchers I.V. Ershova, O.O. Podolyak and A.V. Danilov described a methodology for evaluating the effectiveness of the introduction of robotic complexes (RTK) in the conditions of their growing use in the long term. The study sample consisted of 10 enterprises that implemented FANUC robotic equipment and have been successfully operating for more than a year. Based on an expert survey of enterprise specialists, a ranked list of efficiency factors was obtained: increased productivity, improved quality, reduced labor costs, elimination of hazardous operations, and production flexibility. A correlation–regression model of the dependence of annual savings on selected factors was constructed. After checking the factors for interdependence, four factors remained in the model: increased productivity (labor costs); reduction of defects (quality improvement); harmfulness of work; and the category of work before implementation. The comparison showed that according to expert estimates, the main factor is “productivity growth”, however, calculations showed that the factor “reduction of marriage” comes first. The greatest efficiency of RTK is provided in cases when there is a need to reduce the level of marriage. The authors found that on routine simple operations, marriage can be reduced by two times or more [19]. This technique can be adapted to evaluate agricultural robots, in our case—for use in horticulture. 

In his dissertation, researcher E.A. Skvortsov [20] describes the methodology for substantiating the feasibility of introducing robotics and the methodology for evaluating the effectiveness of its use in agricultural organizations. The author classified agricultural robotics by branches of application: animal husbandry, crop production, auxiliary production, and robotics in crop production by types of work performed: sowing crops, treating plants with pesticides, picking fruits and vegetables, caring for vineyards and horticulture trees, etc. The author identified and systematized the main factors influencing the introduction and use of robotics in agricultural organizations: internal (financial condition of the organization, levels of moral and physical wear of equipment, availability of personnel capable of mastering and servicing robotics, etc.) and external (the price level of agricultural robotics compared to traditional technology, the level of competition among agricultural organizations, infrastructure development, etc.). This made it possible to reduce the influence of factors preventing the introduction of this technique, to increase the efficiency of its use. The principles of the introduction and use of robotics in agricultural organizations were highlighted [20]: priority, quality, complexity, environmental friendliness, economy, efficiency, and safety of use.

This scientific work is of interest, however, only economic factorsweare included in the methodology for evaluating the effectiveness of the use of robotics, while technical factors were not mentioned.

The study by the Cypriot author G. Adamides [21] presented the application of the heuristic evaluation method to test the usability of human-robot interaction systems (HRI) using the example of a semi-autonomous agricultural robot sprayer for vineyards. The following methods were used to design a robot control system: architecture and scalability of the platform, error prevention and recovery, visual design, information presentation, awareness of the robot’s condition, efficiency and effectiveness of interaction, awareness of the robot about the environment, and cognitive factors. In each evaluation study, usability problems were identified and specific proposals for improving usability HRI were documented. In each iteration of the design, fewer usability issues were identified. The author conducted additional experiments that will focus on specific tasks, such as comparing different spraying methods (for example, using a robotic manipulator) and estimating the amount of chemicals saved for spraying, in addition to other gardening tasks to which robotics can be applied. This article is of interest but has a narrow focus—the interaction of a human and a robotic system.

Many authors of publications have investigated the autonomy of robots. The main component of automation in agriculture is autonomous navigation. Currently, extensive research is underway on the use of unmanned automated platforms (UGV) in horticulture. They are used for pruning, weed and disease control, and harvesting. Efficient and high-quality execution of the listed operations is possible if the following conditions are met: autonomous navigation for complex environments; fast operation without damage; and target detection for complex backgrounds. Early navigation systems in agricultural areas used a camera as a sensor and were based on computer vision methods (Santosh A. Hiremath, Gerie W.A.M. van der Heijden et al.) [22]. Navigation, guidance, and transportation included three levels of autonomy: conventional steering, operator-controlled or automatic system (under the control of GO), and a fully autonomous system. Navigation and guidance can be the main task of the system, for example, transporting the crop from the field to the packaging shop, or be an auxiliary task that allows the system to perform its main task, for example, an auxiliary task of spraying or transporting the robot from tree to tree during the harvesting process. Automatic control has been the most active area of research throughout the history of automation of agricultural machinery (Hagras, H.; Colley, M. et al.) [23]. The available systems are based on two main approaches. In the first case, the platform (ground robot) follows a predetermined path based on data from either local Positioning system (LPS) stations or global positioning system (GPS) satellites (Lipinski, A.J., Markowski et al.) [24]. This approach is technically simple, but its disadvantage is the inability to respond to unexpected changes or events in the field (Stentz, A., Dima, C. et al.) [25]. 

With the second approach described by the authors Astrand, B., & Baerveldt, A. J. [26], Bak, T., & Jakobsen, H. [27] the robot operates relative to the sowing line, for example, along a row of plants, or the boundary between plowed and untilled soil or between cut and standing feed, using a sensor system, usually machine vision. This approach allows the robot to adapt its work to individual plants, as they change over time, but it is usually considered that it is technically more difficult to determine the culture line than to follow a certain path [27]. The development of a robotic ground platform that can move autonomously in the changing and dynamic conditions of the external agricultural environment is a complex and difficult task, but it is an important operation for any intelligent agricultural machine [23].

Automatic steering systems for tractors with LPS or GPS control offer farmers the opportunity to: reduce operating costs and increase productivity and profitability (Rovira-Más, F., Chatterjee, I., & Saiz-Rubio, V.) [28]. 

The cconomic benefits include: reduction of overlaps or omissions during fertilization and pesticides, increased timeliness of operations by providing a 24-h work schedule and management in conditions of limited visibility, improved accuracy of water and water. fertilization based on measurements and mapping of plant needs, as well as more effective implementation of accurate farming methods (Bergtold, J.S., Raper, R.L., & Schwab, E.B.) [29]. Authors from France Thuilot, B., Cariou, C. et al. [30] and Japan Nagasaka, Y., Umeda, N., et al. [31] developed an automatic guidance system based on a single RTK-GPS., to guide the tractor along pre-recorded routes. 

The tractor course was obtained by American researchers from the University of North Carolina Welch, G., & Bishop, G. in accordance with the reconstruction of the Kalman state [32], and a nonlinear control law independent of speed was developed.

Although modern navigation systems for agricultural vehicles rely on GPS as the main sensor for steering, an alternative method is still required in cases such as horticulture, where the crown of trees blocks satellite signals from a GPS receiver or [33].

Currently, robotic autonomous platforms are widely used to perform various technological agricultural operations. Research and development in the field of robotic harvesting began in the 1980s, when Japan, the Netherlands and the USA were the pioneer countries.

The efficiency of performing technological operations in gardens by robotic platforms (robots) largely depends on the equipment and sensors, as well as on how well they can perceive the environment in which they move, especially if they move independently, without relying on the intervention of a human operator. The development of autonomous driving is closely related to the ability to interpret and analyze information coming from sensors or combinations of sensors of different types (day and night vision camera, LiDAR, millimeter/ultrasonic radar, etc.). Such sensor combinations are characterized by various optimal operating ranges and allow collecting information related to different sizes of their environment (Andžans, M., Berzinš, J. et al.) [34].

Ukrainian researchers T. Lendiel, I. Bolbot et al. [35] developed a mobile robot with optical sensors for remote assessment of the state of plants and atmospheric parameters in the industrial greenhouse of PJSC “Greenhouse Plant” (“Teplychnyi”), of Kyiv region, Brovary district, village Kalinovka. The algorithm and process of moving a mobile robot in a greenhouse, where its movement is provided by color marks, were described. A non-contact method for assessing the state of plants (the formation of the number of flowers in an inflorescence, the number of fruits on a branch, the average weight and ripeness of the fruit, weight gain) was carried out using wavelet analysis. In this case, each image obtained using a video camera located on a mobile robot is decomposed into wave functions. Training was conducted to gain experience of trial and error of the robot route. It was determined that as experience was gained, the number of unsuccessful attempts and travel time decreased, and the number of incentives received increased. We believe that this movement algorithm does not sufficiently ensure the autonomy of the robot, since the mathematical apparatus is based on the clustering of greenhouse sections. In addition, in the greenhouse it is possible to use the rail robot described by the authors, while in the garden a wheeled platform is needed, the movement of which we plan to describe using differential equations in the X, Y coordinate system.

This article proposes an approach to creating a motion control system for an autonomous robotic wheeled platform based on inertial and satellite navigation and calculation of the traversed path, which will allow it to move in an apple horticulture and automatically perform various technological operations, such as fertilization, control of growth and diseases, harvesting of fruits.

## 3. Materials and Methods

### 3.1. Construction of a Robotic Harvesting Platform in the Horticultural

To conduct field studies of the control system, a developed wheeled robotic platform with a modular design with two running axles was used. The platform was designed to perform various technological operations in industrial garden plantings. The robotic platform consists of an X-shaped frame with four racks, an energy source unit, and an electric drive transmission. Four LPD3806-600BM-G5-24C incremental encoders fixed on the wheel rotation axes and one Autonics EP50S8-1024-1R-P-24 absolute encoder fixed on the steering axis (Figure 2) were used to control the circumferential speed, slipping, angle of rotation and sliding of the wheels.

The movement of the platform is carried out by means of coaxial cylindrical gear motors Transtecno ECMG600-033U (Transtecno SRL, Anzola Emilia BO, Italy) mounted on the rear axle. The power supply system of the robotic platform included a single-phase gasoline power plant with an electric starter LIFAN S-PRO 5500, which was equipped with a system for automatic recharging of lithium-iron-phosphate batteries LiFePO4 24 V 105 Ah while reducing their charge by 20%. The power supply system ensured uninterrupted operation of the platform during an 8-h work shift. For automated steering of the platform, a worm-type steering gear with a Transtecno EC100.120.66 DC electric motor (Transtecno SRL, Anzola Emilia BO, Italy) with a high starting torque was used. A robotic device (Figure 3) was developed to collect apple fruits.

The robotic device has three degrees of freedom, consists of a base, a lower and upper arm, a rotary rack with a gear and a grip. The drives used are a stepper motor with an angle sensor, upper and lower arm displacement actuators CAHB-22E (ERMEC, SL, Barcelona, Spain), boom extension actuator CALA 36A (SKF, Bucharest, Romania), compression and unclenching actuators Wallstech 30 (HK Wallstech Co., Ltd., Hong Kong, China). To control the position of the links when moving in horizontal and vertical planes, the extension of the boom and the rotation of the rack around its axis, Holzer P3022 magnetic (Shenzhen, China) angle sensors and KTR 25 (Chendu, China) linear motion sensors were used. To transport the fruits from the grip to the box, a polyurethane sleeve with a PVC spiral was used, driven by a 24 V 16 A centrifugal extractor. The STM32F207ZGT6 microcontroller (STMicroelectronics, Geneva, Switzerland) is used to control and monitor the positions of the links of the robotic device. The maximum capture reach of the robotic device is 1.5 m, the maximum load capacity at the end of the boom is 0.5 kg, the maximum angle of rotation of the rack around its axis is 270 degrees.

### 3.2. Hardware and Software for Navigation and Motion Control of the Robotic Platform

Possible options for using the navigation and control equipment of the platform are shown in the Figure 4.

In the simplest version of the control system of the robotic platform, navigation is used using the method of calculating the traversed path. For this purpose, low–level sensors that are part of the platform’s control mechanisms are used—sensors for the position of the steering rack, the speed of rotation of the driving wheels, and others. Using well-known algorithms, it is possible to calculate in which direction and by what distance the robotic platform will shift relative to the starting point, knowing the angle of rotation of the wheels and the number of their revolutions. However, this method is the least accurate because it tends to accumulate error quickly due to inaccuracy of measurements, backlash of mechanical parts and uneven soil surface.

To improve the accuracy and correct the above errors, more accurate navigation systems based on the principles of inertial or satellite navigation should be used. Modern satellite navigation systems (SNS) with real-time kinematic technology have a small antenna; provide positioning accuracy of no more than ten centimeters, especially in combination with a remote control and correction base station. This station is constantly located on the ground within radio visibility and accumulates the coordinates of the robot’s location with a high degree of accuracy. The disadvantage of the SNS is its dependence on not always stable satellite signal. To protect against loss or distortion of the satellite signal, it is recommended to use an inertial navigation system (INS) in conjunction with the SNA. It works on the principle of calculating the path, but already on a specialized group of sensors. Such sensors may include accelerometers, electronic gyroscope, and compass, which allow you to determine the angular and spatial position of a robotic platform with high accuracy by calculating accelerations along each of the axes. Such systems are currently used on UAVs. The disadvantage of such a system is the sensitivity of the sensors to the sharp shocks received when driving over the irregularities of the soil surface.

As a result of the analysis, it was proposed in the basic version of the navigation and control system of the robotic platform to combine SNS, INS and a path calculation system for mutual adjustment relative to each other and redundancy in case of failure of one or more of them. To do this, all the above systems supply the primary processed data to the control unit, which filters, combines data from different channels and forms the final coordinates of the spatial and angular position of the robotic platform in the rows of plantings. It was also proposed to equip the radio platform with an interface that would allow the robotic platform to interact with the operator’s device (laptop, tablet, smartphone, etc.) and the software installed on it. The software, in turn, would allow loading and adjustment of the program and the route of movement in the rows of garden plantings, and would also allow tracking the status of completed technological operations and the performance of on-board systems.

The software of the robotic platform, in accordance with the architecture, was built on a modular basis and consisted of the following main elements (Figure 5).

The software consists of two parts—operator and on-board. The operator part contains a module that provides the construction of the route of movement and setting the task for performing technological operations and a module for tracking and visualizing the status of operations performed and the state of the systems of the automated platform. The on-board part is also divided into two parts: the software for controlling the wheel platform and the software for controlling the working bodies (technological adapters).

The software part of the manipulator control system for fruit removal is divided into functional modules: the apple fruit identification module, the manipulator link position control module, the drive control module, and the capture control module.

These modules were implemented on the basis of the ROS framework and are located on the control computer in the developed image of the Linux-based operating system. Each of the modules was implemented as a software node. 

The apple fruit identification module was developed on the basis of stereo vision, allowing the user to recognize the depth of the image and perform three-dimensional localization by constructing a disparity map (D. Alves de Lima et al., 2014) [36]. 

To determine the degree of ripeness of apple fruits, the HSV palette (Hue, Saturation, Value) was used. The color ranges (tones) of ripe and unripe fruit of various apple varieties were used. After specifying the color ranges, a mask was created to overlay the images. Based on the data obtained, a decision is made regarding the ripeness of the fruit on the basis of which it was eaten (D. Khort, A. Kutyrev et al., 2020) [37]. Using the developed stereo camera (Figure 6) allows the user to solve the problem of contact of two or more apples on the crown of the tree. When the stereo camera detects that the distance between two apples is less than or equal to the diameter of one apple, the control system sends a command to the manipulator to eat the apple that is closer first, and then the next apple is selected. This strategy allows the user to successfully harvest apples.

A stereo camera based on two Full-HD resolution webcams was used. The camera was connected via USB cables to a system where an image receiving and processing node was implemented for the ROS framework.

### 3.3. Mathematical Modeling of Robotic Platform Navigation

As a result of the analysis of existing control systems, a model of the movement of a wheeled robotic platform adapted to the conditions of an industrial horticulture was proposed. It is known that in such a garden, trees are planted in rows with a set planting interval. The robotic platform must move in the aisle along a row of trees, making U-turns at the end of the row to enter the next one. Accordingly, the route of the robotic platform can be described by typical trajectories consisting of sections of straight lines and arcs of circles of constant radius. This makes it possible to reduce the movement of the robotic platform to the main typical trajectories (Figure 7).

Turn No. 1 is used when entering the adjacent row, turn No. 2 is used when the width of the turning lane is limited, turn No. 3 is used when the turning radius of the robotic platform is insufficient to enter the adjacent row, turn No. 4 is used when moving to a specific row or completing a technological operation.

The robotic platform was implemented on a four–wheeled chassis, two of which are driving wheels, and two are rotary wheels. In order to implement the above maneuvers, it was necessary to build a mathematical model of the movement of the robotic platform on them. For this purpose, the symbols of the platform characteristics and spatial-angular positions were introduced (Figure 8).

Thus, the high-level controls will be the turning radius *R*, the arc length of the circle performed ∆*φ* and the speed of movement *V*. As can be seen from Figure 4, the length of the arc coincides with the change in the course of the robotic platform when turning, therefore it is indicated by the same value. The operator, when forming the route of the garden detour, can set these values, or they can be calculated from the trajectory of movement drawn by the operator (Figure 9).

To implement a given rotation, it is necessary to introduce low-level controls, which will be the angle of rotation of the front wheels θ and the angular rotation speed of the rear wheels ω. Negative speed will indicate the movement of the automated platform in reverse.

In order to perform a rotation of the required radius, it is necessary to turn the wheels by an angle θ and rotate with an angular velocity ω, which are calculated by the formulas:(1)θ=±asin(LR)ω=±VR

The signs “±” here and further indicate different directions of movement. It is necessary to move until the azimuth of the robotic platform changes by the angle ∆*φ*, found by the formula:(2)$$\varphi =\varphi_{0}\;\pm \;\Delta \varphi$$

In this case, using the parametric notation of the circle equation in polar coordinates, we find that the robotic platform will move to a point:(3)$$x=x_{0}\pm R\cdot \cos\Delta \varphi \;y=y_{0}\pm R\cdot \sin\Delta \varphi$$

Aiming the turn step ∆*φ* to the minimum limit, we obtain a system of differential equations describing the movement of a wheeled robot along the arc of a circle:(4)x′=cosφy′=sinφφ′=VR∗

If we assume that the straight line is an arc of a circle of infinite radius, then this system is also suitable for describing the movement of a robotic board shape along a section of a straight line:(5)$$x^{\prime }=V\cdot \cos\varphi \;y^{\prime }=V\cdot \sin\varphi \;\varphi^{\prime }=0$$

Thus, using the proposed differential equations of motion of a wheeled robotic platform, it is possible to translate complex motion trajectories drawn by the operator into a set of controls (*V, R*, ∆*φ*) (Figure 10).

The developed mathematical model can be used even if the robotic platform has only a basic path calculation system (Figure 4). When installing SNS and/or INS on the robot, the measured current coordinates of the robot (*x*, *y*, *φ*) are compared with the planned coordinates. In case of low-level control deviations (θ, ω), they are corrected by introducing additional terms for stabilizing movement along the route.

The presented method makes it possible to implement a program for automatic movement of a robotic platform along a industrial horticulture with the use of a minimum set of sensors, significantly reducing the load on the processor and memory of onboard computers.

### 3.4. Expert Assessment of the Process of Robotization of the Technological Operation of Harvesting Apples

To select the evaluation factor determining the effectiveness of the robotization process of the technological operation of harvesting apples, an expert assessment was carried out. To do this, a table of individual indicators (factors) was formed with an indication of the rank (Figure 11). Rows with the highest ranking are marked with colors (red, blue and green).

The expert analysis program consisted in selecting the main factors determining the final effectiveness from 18 factors with the help of a group of 30 experts. Each expert ranks individual indicators of the effectiveness of robotics by significance. He puts the most significant, in his opinion, in the first place, and the least significant single indicator in the last place. Each expert can add his own additional indicators to the proposed ranked series, which, in his opinion, also have an impact on the effectiveness of the robotization of the process of carrying out the technological operation of harvesting apples.

The reliability of the results of the survey of experts to determine the ranks is determined by the qualifications of experts and the number of experts. The group should not consist of representatives of the same specialty, since in this case the ranks assigned by them will be the same. A decrease in the number of experts leads to a decrease in the accuracy of the result, since the assessment of each of the experts has an excessive influence on the group assessment. But even with a very large number of experts, it becomes more difficult to identify their agreed opinion due to the reduction in the role of those judgments that, although they differ from the opinions of the majority, are not always proved to be erroneous. Figure 12 shows the resulting graph that characterizes the relationship between the number of experts used in the group and the average group error of their opinions. As can be seen from the graph, the minimum required number of experts should be at least 30 people to increase the accuracy of the result.

Experts from around the world–candidates of technical sciences, doctors of Sciences, members of national academies of sciences in the field of robotics and agricultural mechanization were selected as experts.

To select the evaluation factor determining the effectiveness of robotization and the developed control system, the method of calculating the coefficient of concordance (the method of expert analysis) was applied as a measure of the consistency of a group of experts for each group of factors.

The concordance coefficient, which establishes the level of consistency of the opinions of the expert group on the importance of factors in accordance with the task, is determined by the formula:(6)$$W=\frac{\left(12\times S\right)}{m^{2}}\times \left(n^{3}-n\right)$$
where *S*—the sum of the squares of the difference between the sum of the ranks assigned by all experts to each factor and the average value of the sums of the ranks of all factors; n is the number of factors, m is the number of experts participating in expert analysis.
(7)$$S=\frac{\sum_{j=1}^{n}{\left(\sum_{j=1}^{m}R_{ij}\right)}^{2}-{\left(\sum_{i=1}^{n}\sum_{j=1}^{m}R_{ij}\right)}^{2}}{n}$$
where *R_ij_*—a row in the group after ranking by importance. To identify the consistency of expert opinions on several factors at the same time, it is necessary to find the Kendall concordance coefficient (multiple rank correlation coefficient). The Pearson consistency criterion for determining the significance of the concordance coefficient is found by the following formula:(8)$$\chi^{2}=\frac{12\times S}{m}\times \frac{n}{n+1}=n\times \left(m-1\right)\cdot W\;W=0.13\;\chi^{2}=66.4\;\chi_{табл}^{2}=27.5$$

The analysis of the data obtained to determine the priority factors affecting the effectiveness of robotics allowed us to determine the degree of consistency of expert opinions. A weak degree of consistency of expert opinions has been revealed. However, the calculated *χ*^2^ after comparison with the table value (the number of degrees of freedom *K* = 17, the given significance level α = 0.05) showed that the calculated *W*—value is not random and can be used in further studies.

The analysis of the results of the expert assessment showed that three factors are the most significant:The degree of autonomy of work;Positioning accuracy;Recognition accuracy.

The main results of the expert assessment of the priority of factors are presented in Figure 13.

### 3.5. Methodology for Measuring Indicators of an Automatic Robotic Platform

To assess the level of autonomy of the technological operation, data on the state of the environmental area where the work is being carried out, technical characteristics of the robotic platform, data on the operation (task) being performed and an operation execution plan are required. Based on the data obtained (single indicators of local autonomy), an assessment is carried out of the possibility of performing a planned task with the calculation of a generalized indicator (Figure 14).

To calculate the generalized indicator of the local autonomy of the robotic platform, the methodology was used [16]. The level of autonomy of the planned task is determined by the formula:(9)$$F_{a}=k^{*}\times F^{*}+k_{i}\times F_{i}+k_{en}\times F_{en}$$
where $k^{*},\;k_{i},\;k_{en}$—coefficients of relative importance (weight) that influence the generalized indicator of the local autonomy of the robotic platform.

Determination of the level of the possibility of autonomous fulfillment of the task (carrying out the technological operation of harvesting apple fruits) $F^{*}$ was carried out according to the formula:(10)$$F^{*}=\frac{\sum_{i=1}^{n}N_{i}^{a}\times F_{i}^{m}}{\sum_{i=1}^{n}F_{i}^{m}}$$
where $N_{i}^{a}$—evaluation characteristics of the possibility of performing this stage of the task, $F_{i}^{m}$—significance of this stage of the planned task, n—number of stages of the scheduled task.

The level of the ability to perform the task autonomously reaches a maximum of $F^{*}$ = 1 if all the stages of the scheduled task can be performed by the robotic platform in completely autonomous mode, $F^{*}$ = 0 if the robotic platform cannot perform the task stages autonomously or only with the remote control (remote control). If the calculated value of *F** takes intermediate values, then part of the stage of the planned task can be performed autonomously, and the second part of the task requires the use of a human operator or remote control.

To calculate the *F_i_* time spent on the completion of all stages of the scheduled task, the formula was used:(11)$$F_{i}=F_{i^{*}}\times \overset{m}{\underset{i=1}{\prod }}k_{i}^{c_{t}}\times p_{i}$$
where $F_{i^{*}}$—is the estimated characteristic of the time required to complete the task stages without taking into account the probability of occurrence of various factors, $p_{i}$—is the probability coefficient of an event leading to the addition of a correction factor, $k_{i}^{c_{t}}$—is a correction coefficient that takes into account the time spent on emerging events.

The obtained values of the correction coefficients obtained experimentally are compared with the allowable time to perform the technological operation.

where $F_{i^{*}}$—stimated task completion time without taking into account probabilistic factors, $p_{i}$—the probability of the occurrence of an event leading to the manifestation of the correction factor, $k_{i}^{c_{t}}$—correction factors that take into account the impact of the event on the task execution time. The values of the correction coefficients are determined empirically. Then there is a comparison with the time allowed to complete the task.

The formula is used to calculate the energy spent by $F_{en}$ on the completion of all stages of the planned task:(12)$$F_{en}=F_{en}\times \overset{m}{\underset{i=1}{\prod }}k_{i}^{c_{t}}\times p_{i}$$
where $F_{en^{*}}$—estimated characteristic of the energy costs of a robotic platform for performing all stages of the task without taking into account the probability of occurrence of various factors, $p_{i}$—probability coefficient of occurrence of an event leading to the addition of a correction factor, $k_{i}^{c_{t}}$—a correction factor that takes into account the impact of the emerging event on the total energy costs for completing the stages of the task.

The obtained values of the correction coefficients obtained experimentally are compared with the energy reserves available to the robotic platform for performing all stages of the planned task.

When developing robotic platforms for performing various technological operations, the autonomy of the task is one of the main tasks. Robotic platforms with a high calculated level of autonomy of more than 80% will allow to expand their functional capabilities of performing various operations without human intervention.

## 4. Results

As a result of the conducted research, software was developed to plot the route of movement of the robotic platform, which allowed for the display of the trajectory of movement indicating the accuracy of positioning at each point relative to trees in rows of garden plantings, the speed of movement, and the angle of rotation of the wheels. With the help of the software module, the X, Y, velocity, and azimuth coordinates of the movement are given, as well as the movements of the board-form were visualized along the specified typical turn trajectories in an intensive garden (Figure 15).

The software was developed in the Python programming language. The program interacted with the hardware of the robotic platform. To do this, data transmission via a serial port (COM port) was configured and a program was developed to check data transmission, the result of which was the movement of the robotic platform depending on the data output by the program.

To create a route in the interface of the developed software, the following commands are used:Clicking on the LMB will allow you to plot the route of the technological operation. Red indicates trees (for example, apple trees) that need to be processed;Clicking on the PCM allows you to highlight gaps in the rows of trees (processing of which is not required), they will be highlighted in blackClicking on the LMB in an empty place allows you to mark the stopping points of the robotic platform (in blue).

After the route is created, the trajectory of the detour of the platform breakpoints is visualized. The robotic platform starts moving at the same time, the time, the angle of rotation of the wheels and the speed of their rotation are displayed in the upper left corner. When the program is closed, a file is generated output.xlsx, in which the built route is saved (Figure 16). Green dots indicate trees in rows of plantations, black dots indicate missing trees. The blue color indicates the points of the route of movement, where it is planned to stop near the tree. The red color indicates the trees on which fruit picking operations are currently performed.

After setting up the data transmission, the execution of the route automatically starts, for which data is required from the analog sensor of the steering angle of the left part. To ensure the accuracy of the data obtained, a moving average filter is applied. As a result of the conducted research, the autonomous execution of the specified routes by the robotic platform was realized. Field testing of the developed software as part of a robotic platform was carried out. Experimental studies of the developed control system were carried out in horticulture with the following parameters of plantings: row spacing—3500 mm, trunk spacing—1500 mm, crown width—500–1000 mm, garden type—intensive (Figure 17). The trajectory of movement on the diagram is indicated by a black dotted line. The turning radius when entering the next row did not exceed 3.5 meters. The blue dotted line marks the boundaries of the industrial garden. The yellow dash-dotted line marks the axial lines inside the trajectory of the platform.

The chronology of the route “Apple fruit harvesting with shuttle mode of movement through a row, with a pear-shaped turn” is given in Table 1.

As a result of the analysis of the experimental data obtained, the specified positioning accuracy was confirmed. After pairing and calibration of beacons, the location error, designed with the help of the developed software, does not exceed 3 cm. The results of the field experiment are presented in Table 2.

A fragment of the graph of the dependence of the traveled path on the deviation of the robotic platform from the centerline of the aisle is shown in Figure 18.

The maximum and minimum values of deviation from the inter-row axis are established when the robotic platform moves along the task map using the developed software and when the robotic platform moves along the task map using satellite and inertial navigation systems.

The maximum deviations when moving on the task map using SNS and INS decreased by 22.6%. The standard deviation of the obtained data when moving on the task map using SNS and INS decreased by 64.2% compared to movement using the task map without navigation systems. A generalized indicator of local autonomy has been found:$$F_{a}=\left(k^{*}\cdot F^{*}+k_{i}\cdot F_{i}+k_{en}\cdot F_{en}\right)\cdot 100\%=\left(0.8\cdot 0.65+0.9\cdot 0.259+0.9\cdot 0.125\right)\cdot 100\%=87.5\%$$

It was found that when performing a technological harvesting operation using a robotic platform, the indicator of local autonomy was 87.5%.

## 5. Discussion

Analysis of the results of the conducted research has shown that the proposed indicators are local and cannot solve all the issues related to the description of robot autonomy; however, they allow us to assess the applicability of a particular type of robotic platform for autonomous task execution using various motion control systems. The results allowed us to make a comparison of the preference for using a particular control system to perform a task.

The main interrelations of the components of the developed robotic platform with the level of autonomy are revealed. The analysis showed that the most effective means to expand the functionality of the robotic platform is the use of integrated processing of onboard data: by the level of identification of plant parameters, by the level of identification of environmental parameters, by the level of identification of planning parameters of the robotic platform, by the level of identification of the harvesting object (degree of ripeness, commercial qualities), by the level of completion identification. The structure of the method of hierarchical complex processing of onboard data was developed (Figure 19).

The main advantage of the developed structure of hierarchical complex data processing is the possibility of widespread introduction of feedbacks into the system structure, since its feedback can significantly increase the intelligence of technical systems. The visibility of the reflection of the hierarchy of processes occurring in the information structure of the robotic platform allows users to have a clear and visual structure and correctly describe and trace all the patterns of a complex system. The structure can be recommended for use as part of robot control systems to increase their autonomy and expand functionality.

The calculation of autonomy is given to perform one of the most time-consuming operations in industrial gardening—harvesting apples while moving in rows of plantings. The value of the autonomy indicator when using this technique for other operations will be higher. This is due to the use of working bodies with a smaller set of connected sensors and a simpler control system.

In recent years, a promising direction is the use of ammonia as an alternative fuel that can replace existing fossil fuels. A hydrogen carrier, zero carbon emissions, liquid unlike hydrogen, and can be produced using renewable resources, making ammonia the future green fuel for internal combustion engines. Ammonia can become a convenient energy carrier and the basis for efficient energy storage (battery pack) used on a robotic platform to increase its autonomy.

The American technology start-up company Amogy based on the John Deere 6195M model created the world’s first tractor powered by liquid ammonia [38]. It was demonstrated at the Center for Advanced Technology and Energy at Stony Brook University in New York (USA). As part of its run-in, the equipment confirmed the status of a car with zero carbon dioxide emissions, since nitrogen and water were released during the combustion of fuel. The Amogy system consists of a standard liquid ammonia tank and specialized fuel cells.

The studies of French authors C. Mounaïm-Rousselle and P. Brequigny [39] were carried out on CFR engines and on laboratory stands of single-cylinder engines with stable initial thermal conditions. Results from modern SI combustion chambers have confirmed that a compression ratio of around 10:1 may be sufficient to burn ammonia with little H_2_ (about 5–10% vol.) or even no H_2_ in a “full” load for future hybrid vehicles. or range extension systems.

Nadimi, E., Przybyła, G., Emberson, D. et al. [40] experimentally studied an ammonia biodiesel dual-fuel engine with spark ignition (SI) for subsequent use in agricultural machinery. The single-cylinder diesel engine has been modified to inject ammonia into the intake manifold, and then a trial dose of biodiesel is injected into the cylinder to initiate combustion of the premixed ammonia-air mixture. The results showed that 69.4% of the energy consumed by biodiesel can be replaced by ammonia but increasing the mass flow rate of ammonia slightly reduces the thermal efficiency of the brakes. In addition, an increase in the ammonia load contribution significantly reduced CO_2_, CO and HC emissions, but increased NO emissions.

The use of ammonia as fuel will allow the joint use of three energy sources of a gasoline gasoline generator, batteries and an ammonia engine, which will avoid working in low-load mode, as well as realize kinetic energy recovery and distribution of energy consumption in various operations, increasing the fuel efficiency of the power plant and battery life without recharging.

## 6. Prospects for Further Research

In order to increase local autonomy during the apple harvest operation, it is planned to equip a robotic platform with a lifting device for automated capture and transportation of a plastic container (Figure 20).

The group application of a universal robotic platform with an automated lifting device paired with a robot for transporting fruit-filled containers will increase productivity, reduce the cost of container placement, the process of their selection and transportation to the warehouse. 

The modular design of the robotic platform makes it possible to use ammonia energy as a driving force. The developed system includes a tank with liquid ammonia, fixed on the frame of a robotic platform, and fuel cells (Figure 20). The use of ammonia makes it possible to increase the autonomy of the robot, as well as to use the zero level of sights. We plan to conduct experiments on the operation of this robotic platform paid form using the ammonia engine described in [40] in order to increase the efficiency and autonomy of the robot based on an alternative energy source.

In further work, it is also planned to develop an intelligent control system for the equipment of a robotic device for picking fruits, including a neural network for identifying fruits on tree crowns. In addition, it is necessary to predict the risks of investors in business projects associated with the introduction of robotics in agriculture, according to the “European Green Deal” in connection with possible threats in the food market (Faichuk, O.; Voliak, L.; Hutsol, T. et al.) [41] and conduct ecological and economic justification of the introduction of robotic technologies in horticulture. In the future, it is planned to expand the use of the developed autonomous robots to other branches of agriculture, including the automation of production processes in livestock buildings [42], in greenhouse [43], when planting and unloading the energy willow cuttings [44] et al.

## 7. Conclusions

The expert analysis of the evaluation of priority indicators of the technological operation of harvesting apples using a robotic platform of 30 experts allowed us to choose the main factor from 18 factors that determined the effectiveness of the process of carrying out the technological operation of harvesting apples using a robotic platform. It was established that the most significant factor is the autonomy of performing operations.

As a result of the conducted research, a motion control system for an autonomous robotic wheeled platform based on inertial and satellite navigation and calculation of the traversed path was developed. The developed software allows users to design the route of the robotic platform, move in the apple horticulture and automatically perform various technological operations, such as fertilization, control of growth and diseases, and the harvesting of fruits. The developed modular architecture of the control system allows it to be supplemented with the following extensions:-collision avoidance system with people, animals and obstacles based on ultrasonic or laser sensors, for example, car parking radars;-the system of recognition of surrounding objects based on video or infrared camera;-a system for constructing a three-dimensional map of surrounding objects and landscape based on LiDARs;-SLAM system for building a map of the surrounding area based on visual sensors.

The developed platform has significant differences from its analogues: increased ground clearance, high cross-country ability, adaptive control system with various navigation systems, compact design, and low weight to increase its maneuverability in the field and reduce pressure on the soil. The modularity of the design of the robotic platform allows it to be supplemented with electronic components for analyzing the operation of technological modules in order to obtain information about the level of autonomy of the platform and the possibility of increasing it. The method of increasing the level of autonomy of the robotic platform has the possibility of expanding the individual indicators taken into account in more labor-intensive production tasks.

To increase the autonomy of the robotic platform, it is necessary to increase the speed of operation of the robotic device for collecting fruits and the interference protection of the control unit to external electromagnetic influences, replacing it with batteries with a higher capacity.

## Figures and Tables

**Figure 1 sensors-22-08901-f001:**
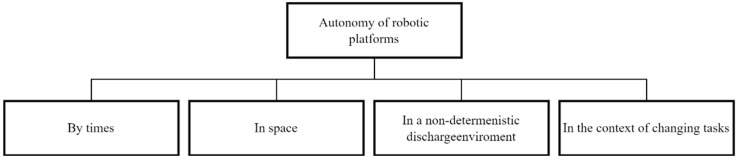
Classification of Autonomous robots.

**Figure 2 sensors-22-08901-f002:**
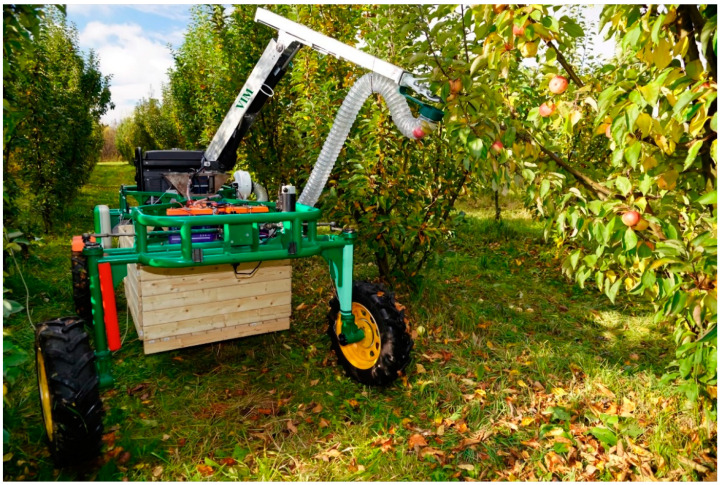
Wheeled robotic platform with a robotic device for fruit removal.

**Figure 3 sensors-22-08901-f003:**
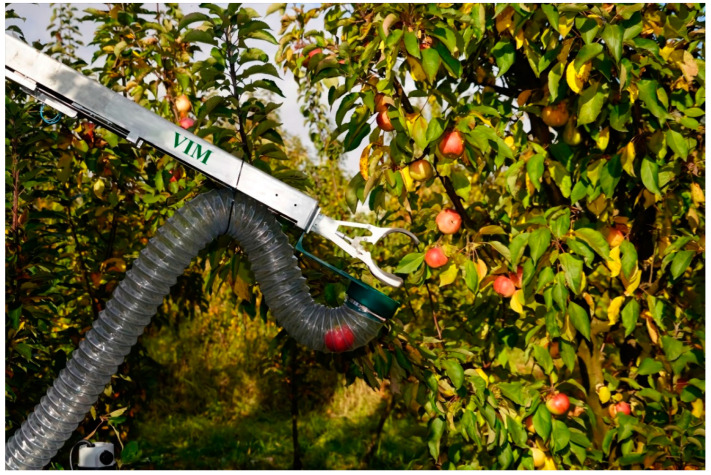
A robotic device mounted on a robotic platform for collecting apple fruits.

**Figure 4 sensors-22-08901-f004:**
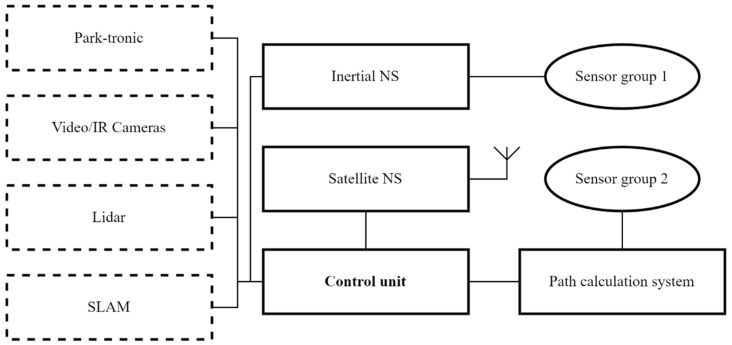
Block diagram of various variants of navigation and control equipment of the robotic platform.

**Figure 5 sensors-22-08901-f005:**
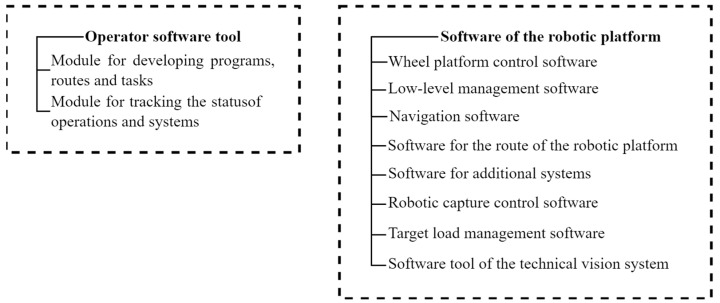
Elements of the robotic platform software.

**Figure 6 sensors-22-08901-f006:**
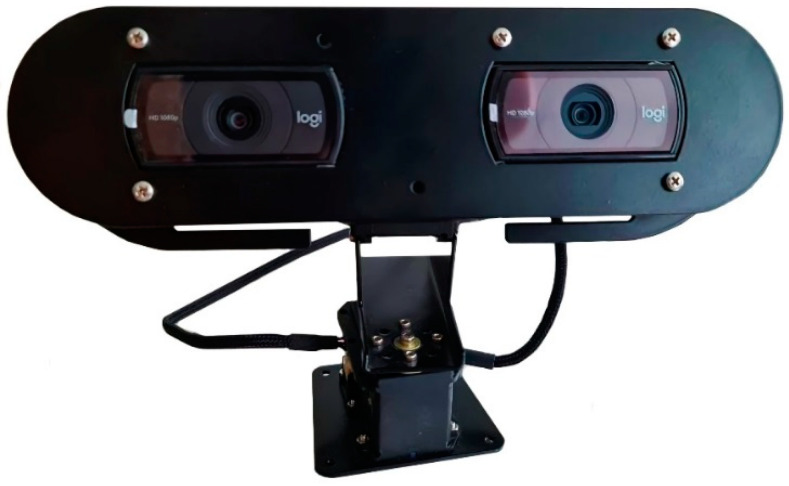
Stereo camera of the apple fruit identification system.

**Figure 7 sensors-22-08901-f007:**
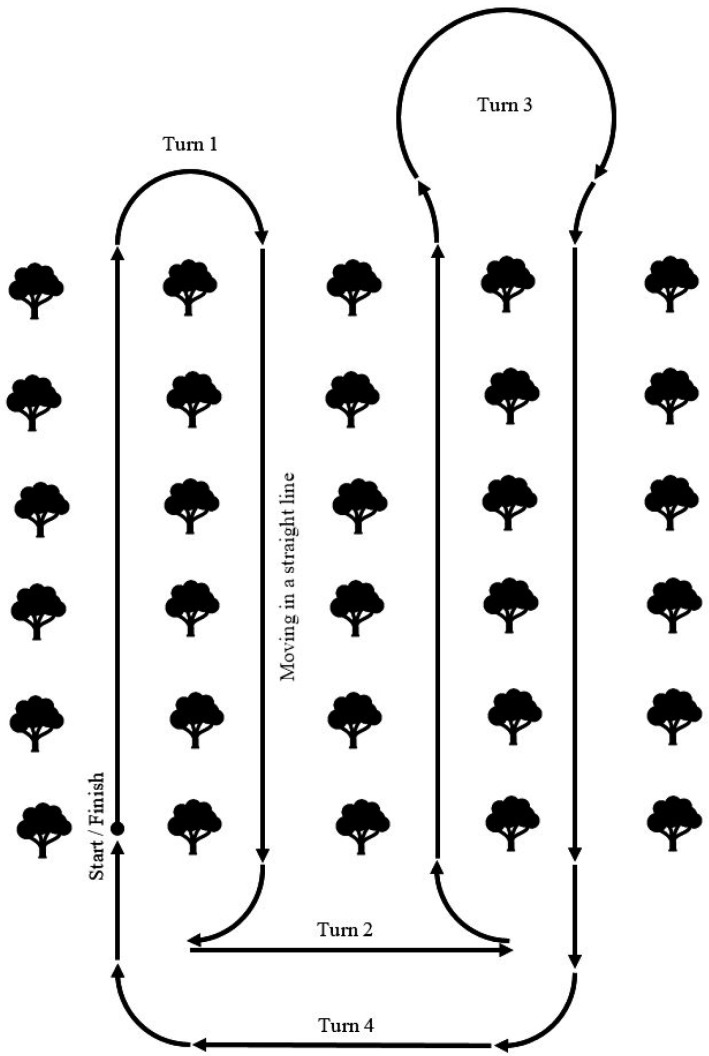
Typical trajectory of a robotic platform.

**Figure 8 sensors-22-08901-f008:**
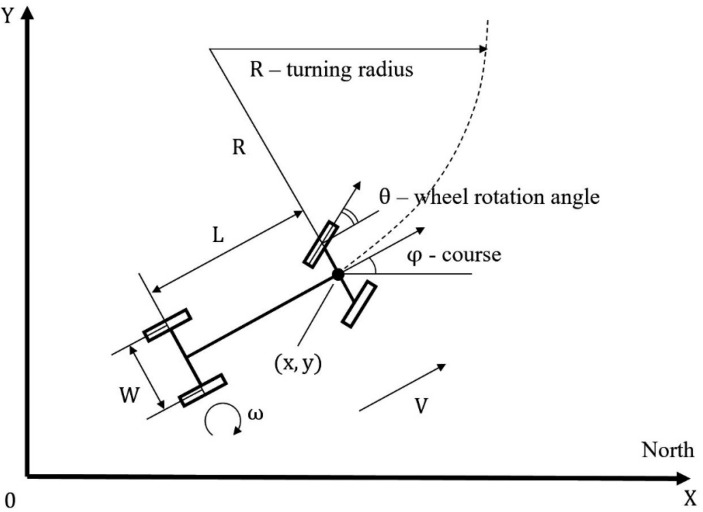
Designations of the spatial-angular position of the robotic platform.

**Figure 9 sensors-22-08901-f009:**
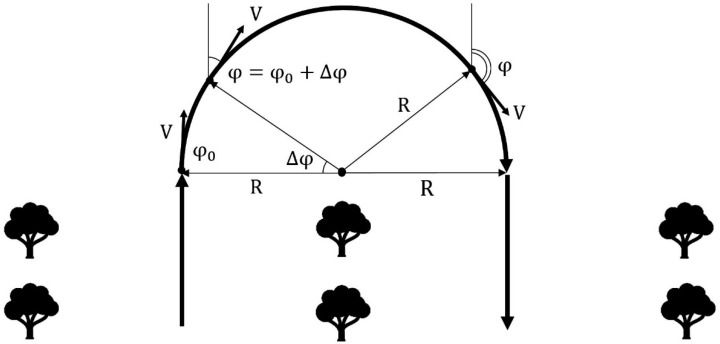
High-level control calculation.

**Figure 10 sensors-22-08901-f010:**
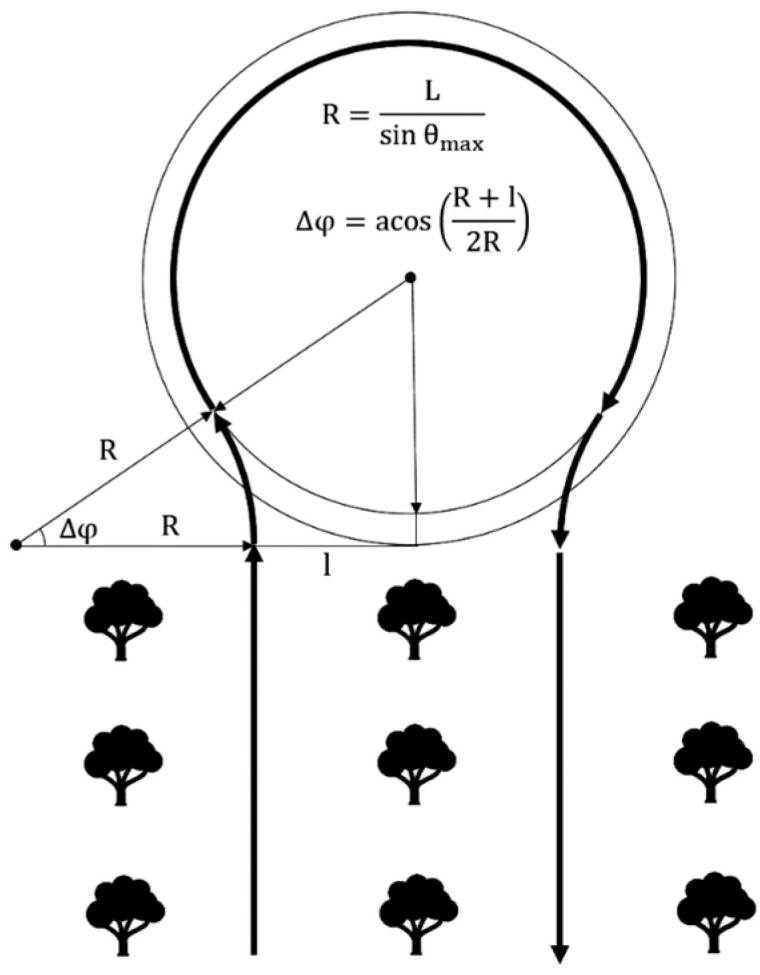
Calculation of controls on a complex trajectory.

**Figure 11 sensors-22-08901-f011:**
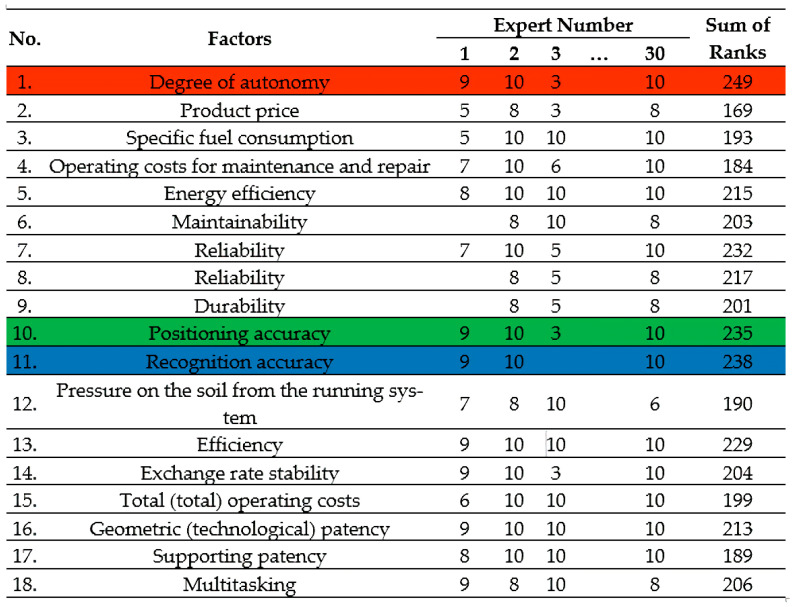
Selected factors and their ranking results.

**Figure 12 sensors-22-08901-f012:**
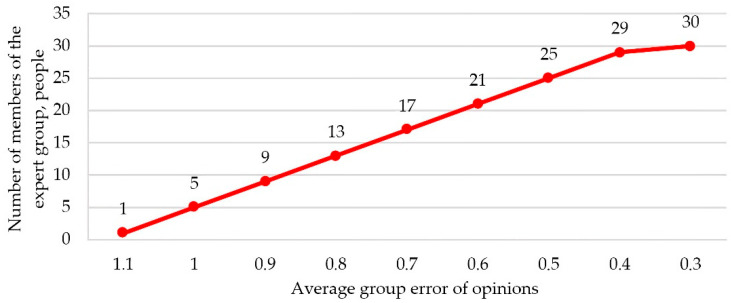
Determination of the minimum required number of experts.

**Figure 13 sensors-22-08901-f013:**
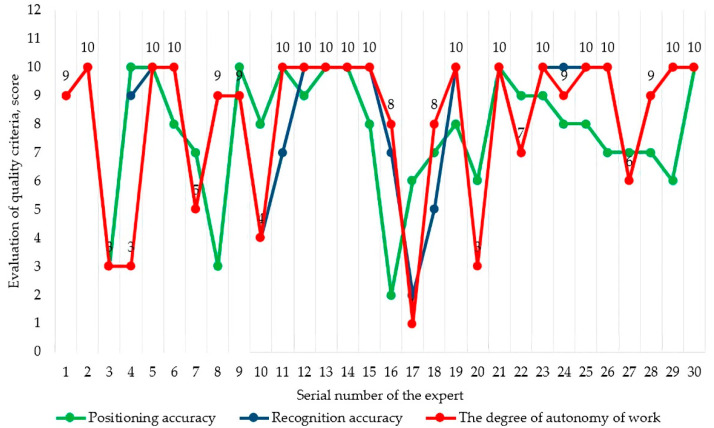
Ranking of priority factors.

**Figure 14 sensors-22-08901-f014:**
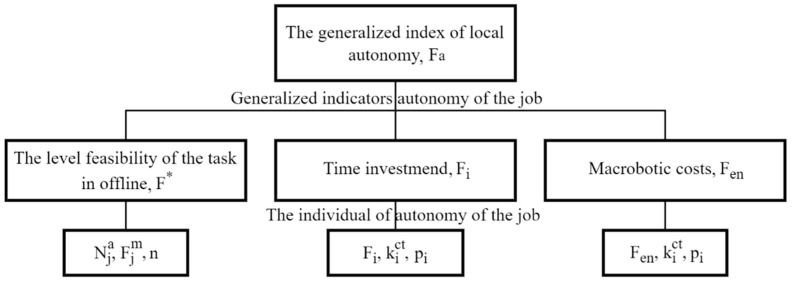
A block diagram of the definition of generalized and individual indicators of the autonomy of performing operations of a robotic platform.

**Figure 15 sensors-22-08901-f015:**
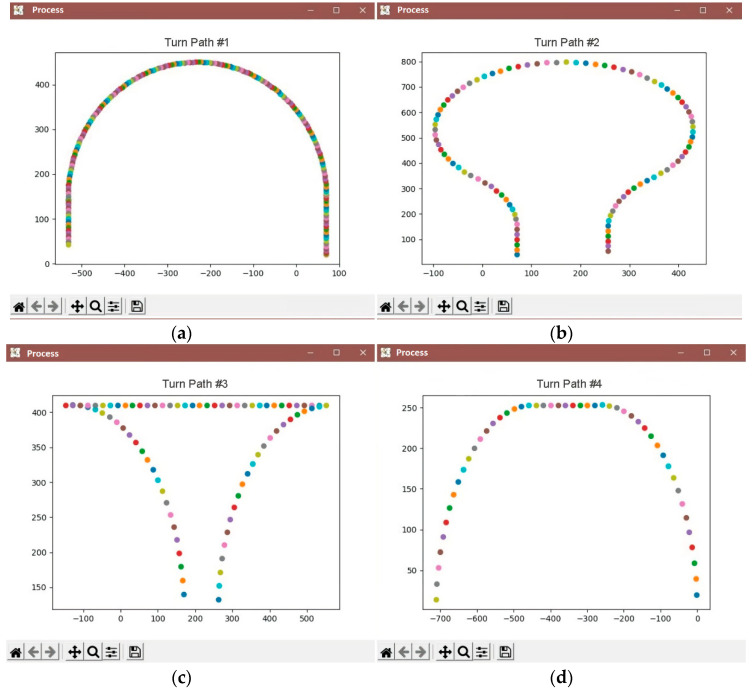
The specified typical trajectories of the platform turn in an intensive garden: a turn through a row (**a**), a turn of a large radius (**b**), a turn using reverse gear (**c**), a turn with a straight line (**d**).

**Figure 16 sensors-22-08901-f016:**
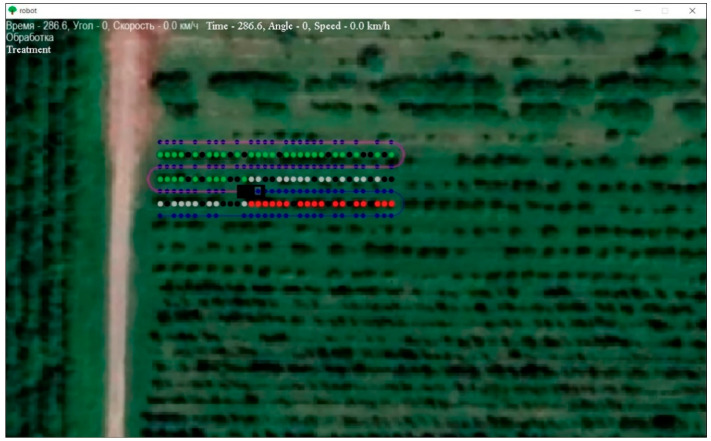
Interface of the software module for building the route of the robotic platform.

**Figure 17 sensors-22-08901-f017:**
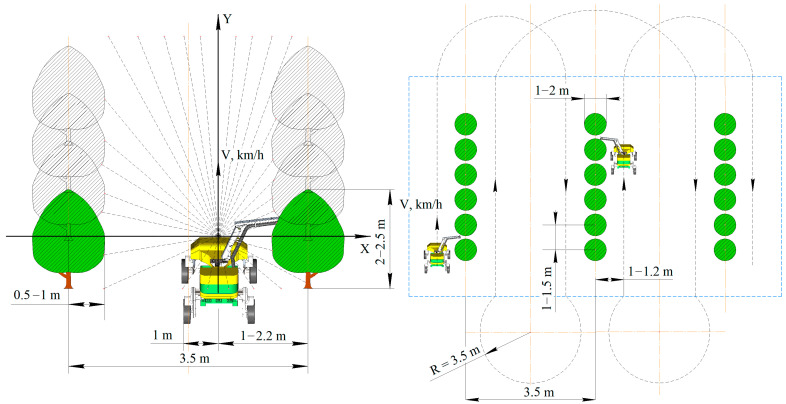
Garden parameters and the choice of the method of movement of the robotic platform.

**Figure 18 sensors-22-08901-f018:**
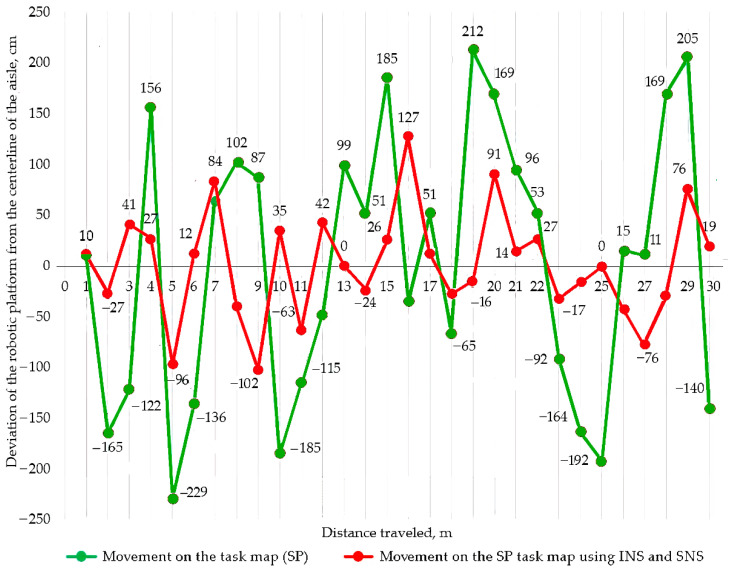
A fragment of the graph of the dependence of the traveled path on the deviation of the robotic platform from the centerline of the aisle.

**Figure 19 sensors-22-08901-f019:**
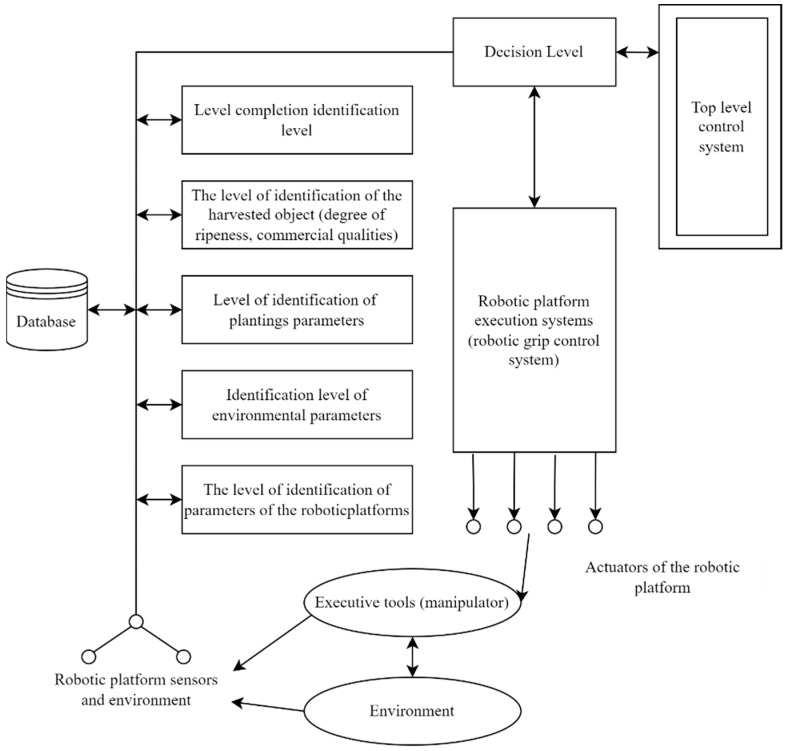
The structure of the method of hierarchical complex processing of on-board data.

**Figure 20 sensors-22-08901-f020:**
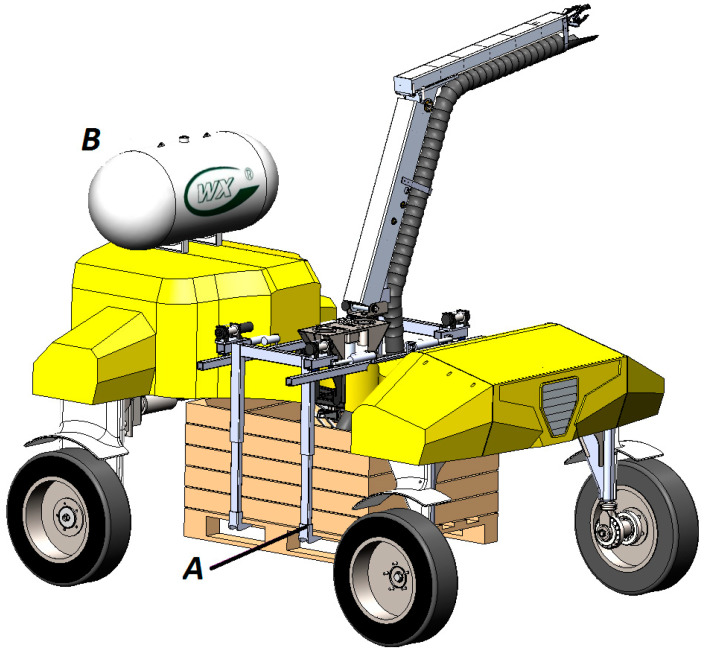
A device for automated capture and transportation of a fruit container (**A**); and tank with liquid ammonia (**B**).

**Table 1 sensors-22-08901-t001:** Chronology of the route “Apple fruit harvesting with shuttle mode of movement through a row, with a pear-shaped turn”.

No.	Type of Movement	Meaning
1	2	3
1.	Program start, start of movement, s	5
2.	Setting the speed of movement along the line of the row, km/h	5
3.	Movement along the line of the row, s	220
4.	Stop near each tree in a row to identify and collect fruits (positioning relative to the trunk of the tree), s	400
5.	Start of moving to adjacent row, s	1
6.	Setting the speed of movement in a turn, km/h	3
7.	The beginning of turning the steering part to the right, s	14
8.	The turn is completed, the movement is straight, along the line of the row, s	220
9.	Setting the speed of movement along the line of the row, km/h	5
10.	Stop near each tree in a row to identify and collect fruits (positioning relative to the trunk of the tree), s	400
11.	The start of the pear-shaped reversal, s	1
12.	Setting the speed when turning, km/h	3
13	The beginning of turning the steering part to the right, s	27
14.	The turn is completed, the movement is straight, along the line of the row, s	220
15.	Repeating the cycle of operations	-
16.	Total, calculation of the duration of the route, s	1508

**Table 2 sensors-22-08901-t002:** Results of the field experiment.

Deviations Minimum, mm	Deviations Maximum, mm	Standard Deviation, mm	Variance over the General Population, mm^2^
When moving on the task map			
−229	212	117.56	17,659
When moving on the map tasks using satellite and inertial navigation systems			
−102	164	42	2828

## Data Availability

Not applicable.

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
