# Peer review of "Robotic Platform for Horticulture: Assessment Methodology and Increasing the Level of Autonomy"

_sensors, 2022, doi:10.3390/s22228901_

Round 1
Reviewer 1 Report
Dear Authors, Highly appreciated for research work in Field Robotics, it is interesting research in the agriculture/ horticulture based societal application
Well contributed for horticulture challenges with robotic platform.
1. Literature survey did well as per problem statement
2. Interesting contributions are
a. Robotic platform for picking apple properly using robotic arms using both lower and upper.
b. Robotic path planning illustrated well for typical trajectories with mathematical models and proper results
c. Robotics is integrated with both software and hardware
d. Fuel challenges are also addressed properly using battery and ammonia air mixture methods.
3. Overall research impacts the horticulture, it is societal application.
Questions to address
1. In process of the apple pick from tree, cameras are used, but which algorithm or software’s used to estimate fruit degree ripeness.
2. How robot analysed the ripeness of fruit with image processing algorithm.
3. How it is differentiating cluster of ripeness fruits are in same bunch.
Kindly provide solutions for above questions with technical details in lines of image processing approaches.
Author Response
Dear reviewer!
Thank you for your comments and positive evaluation of our article. The file contains responses to your comments.

Reviewer 2 Report
The paper develops an autonomous robotic platform that can increase the level of autonomy in the navigation of a robotic platform used in horticulture. The robotic platform is ultimately intended for performing the task of plucking apples in horticulture as well as other tasks. Based on a survey performed using 30 experts in the field, the paper identifies and ranks 18 factors for determination of the efficiency of an autonomous system for horticulture. The three most important factors are found to be: degree of autonomy of work, positioning accuracy and recognition accuracy. The paper describes a robotic platform as well as a system for controlling the movement of the robotic platform using a combination of SNS, INS and path planning. A software module is designed to control the trajectory of the robotic platform and mathematically control how the platform turns when moving from one row to another. The combined use of SNS, INS and path planning improves the accuracy of navigation of the robotic platform.
Strengths:
1. The paper introduces a robotic platform and a system for controlling the autonomous navigation of the platform through an apple orchard using mathematical formulas and differential equations to control the turn trajectories.
2. The system combines the use of SNS, INS and path planning to improve the accuracy of navigation accuracy of the platform instead of using just one of these techniques, each of which suffer from lack of accuracy or from down-times (example: satellite information being unavailable for SNS).
3. Uses a survey from 30 experts in the field to identify and rank important factors for determining the efficiency of an autonomous robotic platform.
Weaknesses:
1. The developed system and platform only involve descriptions about navigation of the platform and do not include a description of how apple plucking is performed autonomously.
2. The work is tested in only one specific environment and its applicability to a different environment or orchard has not been studied. The performance of the system when the garden parameters are varied is not presented.
3. Other factors mentioned in the revisions section.
Suggested revisions:
1. The arm used for harvesting is described but how it can detect an apple on the tree and pick it, is not. The future works mentions using deep learning to identify an apple on the tree. However, how is this done autonomously at present? This part needs to be addressed in the manuscript.
2. Figure 4 mentions SLAM, camera, LIDAR in addition to SNS and INS. However, it seems that these are proposed for future additions and only SNS and INS are used at present. In that case, the diagram either needs to be modified or the caption modified to state this fact.
3. The discussion section mentions that the use of ammonia as a fuel will make the system more autonomous. It is not clear how changing the fuel improves the autonomy of the system. Please clarify this.
4. If the system is indeed focusing on autonomous navigation and not on autonomy in actual apple harvesting in terms of plucking from the tree, that idea needs to be clarified throughout the paper. In that case, this is a pre-cursor to the system for autonomous apple harvesting and implements and builds the navigation module of the task only.
5. Methods section should be broken up into sub-sections to improve readability.
Author Response

(The authors gave the same response as above.)
